# Beyond Photoprotection: The Multifarious Roles of Flavonoids in Plant Terrestrialization

**DOI:** 10.3390/ijms23095284

**Published:** 2022-05-09

**Authors:** Luana Beatriz dos Santos Nascimento, Massimiliano Tattini

**Affiliations:** 1Department of Agriculture, Food, Environment and Forestry (DAGRI), University of Florence, 50019 Sesto Fiorentino, Florence, Italy; 2Institute for Sustainable Plant Protection (IPSP), National Research Council of Italy, 50019 Sesto Fiorentino, Florence, Italy; massimiliano.tattini@ipsp.cnr.it

**Keywords:** arbuscular mycorrhizal, auxin, drought, flavonoids, nutrient scarcity, symbiosis, terrestrialization

## Abstract

Plants evolved an impressive arsenal of multifunctional specialized metabolites to cope with the novel environmental pressures imposed by the terrestrial habitat when moving from water. Here we examine the multifarious roles of flavonoids in plant terrestrialization. We reason on the environmental drivers, other than the increase in UV-B radiation, that were mostly responsible for the rise of flavonoid metabolism and how flavonoids helped plants in land conquest. We are reasonably based on a nutrient-deficiency hypothesis for the replacement of mycosporine-like amino acids, typical of streptophytic algae, with the flavonoid metabolism during the water-to-land transition. We suggest that flavonoids modulated auxin transport and signaling and promoted the symbiosis between plants and fungi (e.g., arbuscular mycorrhizal, AM), a central event for the conquest of land by plants. AM improved the ability of early plants to take up nutrients and water from highly impoverished soils. We offer evidence that flavonoids equipped early land plants with highly versatile “defense compounds”, essential for the new set of abiotic and biotic stressors imposed by the terrestrial environment. We conclude that flavonoids have been multifunctional since the appearance of plants on land, not only acting as UV filters but especially improving both nutrient acquisition and biotic stress defense.

## 1. Introduction

Many evolutionary innovations, i.e., the adaptive traits of living organisms [1], occurred over a short time period when plants moved from water to populate land (approx. 460 mya) [2,3]. Together with other adaptive features, “molecular” innovations were driven by the novel environmental pressures that the early land plants faced in harsh terrestrial habitats. These included dramatic declines in water and nutrient availabilities, large fluctuations in air temperature, and substantial increases in visible light and UV radiation [4,5,6]. However, there is now conclusive evidence that most of the molecular toolkit enabling the conquest of the terrestrial habitat by plants was already present in their ancestors, i.e., members of the Charophyta algae clade (Zygnematophyceae) [4,7,8,9]. This molecular tool kit included a set of novel transcription factors, and genes involved in both phytohormone signaling and in the formation of the cell wall, all of them being closely linked to the emergence of land plants [8]. Therefore, in other words, plants were apparently “terrestrial from the beginning” [10].

For instance, the two- (2D) to three-dimensional (3D) growth transition was a key innovation increasing the plant complexity [11,12] and hence improving their ability to conquest more stressful habitats [13,14,15,16]. This is consistent with the notion that 3D plants bodies display higher resistance against desiccation and UV radiation, ultimately providing protection to both vegetative and reproductive tissues [6,17]. Interestingly, recent findings show that 2D to 3D growth transition mostly involves the phytohormone auxin and its asymmetrical distribution in different plant tissues, driven by the polar auxin transport (PAT) [12,18]. Indeed, PAT was already present in the ancestors of land plants, and it was regulated by a range of transporters, including the PIN-formed (PIN) proteins [19]. Nonetheless, PAT in Charophytes is not as tightly regulated as it is in bryophytes and angiosperms [20,21]. This only in part is due to the increased numbers of PIN proteins detected in more complex land plants. Instead, the evolution of regulatory mechanisms of both PIN activity and PIN-induced PAT likely represented a major developmental innovation responsible for the increased complexity observed from ancestors of land plants up to angiosperms [22,23]. Similar reasoning concerns core components of the ABA signaling pathway, such as SnRK and PP2C proteins, which were already present in land plants’ ancestors [24,25,26]. The complexity and robustness of the ABA signaling network observed in gymnosperms and angiosperms are due to not only the expansion of core components but also to the evolution of several downstream regulatory network components (MYB transcription factors and MAPKs proteins) [27,28,29,30]. The enhanced complexity in this pathway has expanded the function of the ABA signaling up to include the fine tuning of stomata movements, a trait that would allow land plants to later adapt to habitats characterized by severe soil water deficit and excessive sunlight irradiance [30,31].

The outstanding ability of terrestrial plants to adapt to this “ever-changing” new environment has depended not only on the evolution of a pre-existing molecular toolkit but largely on the impressive chemical diversity originated by the rise of a huge number of specialized metabolic pathways [32,33,34]. The pivotal role of specialized metabolism in the successful responses of plants to environmental stressors also depends on the huge numbers of metabolites synthesized by different taxa. It is long known that each specialized metabolite class serves a multiplicity of functions in plant-environment interaction [35], changing with the plant developmental stage and the type and intensity of environmental injuries [36,37,38]. Indeed, these metabolic networks enable plants’ diversification, biotic interactions, and, to a greater extent, the occupation of several niches [34]. Therefore, the multi-functionality represents an effective way to reduce the metabolic cost of specialized metabolite biosynthesis [39], being, indeed, a widespread property of most secondary metabolites [40].

In this brief review, we examine the multi-functionality of flavonoids, the ancient class of secondary metabolites synthesized through a branch-pathway of the general phenylpropanoid metabolism, which is unique to land plants [41,42], in the “plant terrestrialization”. The primary functions of flavonoids during the water-to-land transition are still uncertain [41,42]. This originates from (1) the inherent ability of flavonoids to serve multiple functions in the response of plants to environmental stimuli and (2) the contemporary action of novel environmental stressors plants faced in the terrestrial habitat [37,43,44,45,46,47]. We reason on the environmental drivers that were mostly responsible for the rise of the flavonoid metabolism, a pre-requisite to unveil their primary roles during the early steps of plant terrestrialization.

## 2. A “Revolutionary” Molecular Innovation: The Rise of Flavonoid Metabolism during Water-to-Land Transition of Plants

One of the most intriguing molecular innovations that accompanied the water-to-land transition of plants was the replacement of the mycosporine-like amino acids (MAAs) metabolism, typical of streptophytic algae, by the flavonoid biosynthetic pathway (Figure 1), a common feature of all extant land plant taxa [41,48]. Actually, there is evidence of the occurrence of early genes of the phenylpropanoid pathway, such as those involved in lignin and coumarin biosynthesis, in algal relatives closest to land plants [4,49]. This was of pivotal value for cell wall reinforcement and for the initial steps of both plant vascularization and defense, allowing their successful growth and development on land [50]. Indeed, the localization of phenolic compounds in different cell organelles in comparison to the predominantly cytoplasmatic position of MAAs [51,52] supports the importance of phenolics in the early steps of land conquest. We suggest that the presence of both MAA [53] and phenylpropanoid metabolism [49] in clades sisters of the early land plants was an expensive adaptation strategy, and it is likely due to reminiscence of the evolution of both basal and highly branched streptophytic algae in nutrient-rich habitats. Indeed, it is speculated that early land plants were firstly dependent on MAAs instead of flavonoids as UV protectants [54]. We observe that while flavonoids are merely carbon-based compounds, MAAs have nitrogen atoms in an amino-cyclohexanone or amino-cyclohexenimine core skeleton [48,52] (Figure 1). Therefore, it is conceivable that when plant ancestors moved to land, the MAA metabolism was unfavored and further definitively lost because of the large deprivation of nitrogen (as well as phosphorous) and water in land soils [2,48,55]. This “saving strategy” [56,57] allowed the nitrogen to be fully available for sustaining the growth of early plants in nutrient-poor terrestrial habitats. Indeed, several studies have shown a positive correlation between flavonoids’ biosynthesis and nitrogen depletion [58,59], and a high C/N ratio has been reported to promote the accumulation of flavonols [60]. It is suggested that, despite the biosynthesis of flavonoids requiring high-energy consumption, their preferential accumulation might represent a “power drain valve” in a scenario of higher solar irradiation [61,62], such as that faced by early plants. However, the matter is much more complex than we discuss here. As speculated by Harholt et al. [10], the Streptophyta algae were “terrestrial from the beginning”, which is consistent with the rise of phenylpropanoid metabolism in several of these lineages. Nonetheless, we note that Charophyceae ancestors of land plants first colonized moderately moist habitats near freshwater, then gradually moved into drier lands [56,63].

The replacement of nitrogen-containing MAAs with carbon-based products synthesized by the “late” flavonoid pathway, particularly the flavonol branch, was, therefore, a molecular event of outstanding significance during plant terrestrialization [30,48,64], although this matter has not received the deserved attention in the pertinent literature yet. The nutrient-centered hypothesis for the rise of flavonoid metabolism outlined above is reasonable but also raises a series of concerns that needs deep analysis. For instance: (i) it is necessary to determine which are the benefits associated with flavonoid metabolism in primitive terrestrial plants that were facing nutrient and water-deficient soils; (ii) it moves to the background of the UV-B centered hypothesis, i.e., the increase in UV-B irradiance was the evolutional environmental driver responsible for the rise of the flavonoid metabolism.

## 3. Arbuscular Mycorrhizal Associations: A Central Event in Plant Terrestrialization Aided by Flavonoids

As outlined above, one of the most severe challenges faced by early land plants was the drastic reduction in the availability of water and nutrients [65,66]. Once again, land plant ancestors were already equipped with a toolkit potentially enabling their adaptations to land, which includes their ability to form symbiotic associations with fungi and bacteria [67]. This is intriguingly in line with the findings that nutrient-impoverished-ancient soils are indeed hotspots of plant biodiversity [68,69]. It is also consistent with the recent suggestion that ancestral and modern streptophyte algae could recognize fungi since they have homologous genes of LysM-RLK receptors, through which land plants recognize fungi [67,70,71]. However, even the most branched/complex algae miss other genes required for mycorrhizal association (MA), including those involved in the formation of arbuscules [72]. In fact, mycorrhizal symbiosis is a shared derived character, unique to land plants [67] (Figure 2a), playing a key role in the terrestrialization 450 million years ago [73,74,75]. The current paradigm is that the earliest rootless terrestrial plants coevolved with Glomeromycota arbuscular mycorrhiza fungi that, in exchange for plant photosynthates, enhanced their access to mineral nutrients and water [76,77,78,79]. There are few examples of AM or similar associations in streptophytic algae [71], whereas liverworts, hornworts, and lycophytes have recently been shown to form symbioses with members of two ancient fungal lineages: arbuscular mycorrhizal fungi of the Glomeromycotina and the symbiotic fungi of the Mucoromycotina [74,80]. Even if recent reports have revealed that fungi associations between liverworts and members of the Glomeromycotina are particularly few [81], a limited number of species have been studied so far [82]. The outstanding significance of arbuscular mycorrhiza fungi in assisting not only the water-to-land transition [83] but mainly in the expansive conquest of land by plants is also well exemplified by the observation that more than 80% of plant species form AM [84,85]. Notably, a study conducted with members of the different plant groups revealed that all the gymnosperms surveyed were mycorrhizal, most of them obligate [86], which was attributed to their evolution in nutrient-poor habitats. Moreover, there is compelling evidence that even symbioses between “modern” flowering plants and nitrogen-fixing bacteria (e.g., Rhizobium) must probably have evolved from the ancient AM symbiosis since several of the pathway signals are common between both biological interactions [77,87].

Auxin is a major regulator of plant growth and developmental processes, and it is a central modulator/regulator in AM symbiosis [88] (Figure 2a). Increased auxin level in colonized cells has promoting effects on hyphal branching (possibly by weakening the cell wall) and, consequently, on arbuscule incidence [89,90]. There is also evidence of auxin involvement in the early stages of AM formation, e.g., during pre-symbiotic signal exchange [91], in part through the control of the strigolactone levels [92]. The feedback regulation of the host IAA biosynthesis offers strong support to the tight auxin-AM relationship. This mechanism avoids the excessive auxin accumulation in colonized cells and modulates the subsequent auxin-induced gene expression, both contributing to the phenotypical changes during mycorrhizal colonization [90,93].

The strong dependence of AM on the auxin perception and signaling highlights the importance of endogenous regulators of this hormone, such as the flavonoids [94,95]. Actually, several secondary metabolic pathways are interconnected with phytohormone networks, making most specialized metabolites active in plant growth and development regulation, besides their general metabolic function [34]. Flavonoids have long been reported to be involved in AM formation [96,97,98,99], being active in root colonization, spore germination, hyphal growth, and branching [47,100,101]. Nonetheless, the molecular mechanisms that drive the effects of flavonoids in AM are largely unexplored. It is conceivable that the flavonoid-induction of AM partially involves the regulation of local auxin levels (by acting on its transport and catabolism) [102] and of the level of downstream components of the auxin signaling pathway, as well known to occur in flavonoid-induced nodulation [103,104]. As outlined above, some features of root nodule endosymbiosis have been likely recruited from the more ancient AM symbiosis [105,106], and there is compelling evidence that flavonoids act as both essential signals for the establishment of legume nodulation and prime candidates in AM symbiosis [107]. The observation that flavonoid aglycones are much more active than flavonoid glycosides in the promotion of AM, likely through the inhibition of auxin transport, similarly to what is observed during nodulation [108], further corroborates the idea of a strong relationship between flavonoids, auxins and AM [47].

Overall, this is consistent with the relatively old suggestions that flavonoids play a crucial role as developmental regulators [109,110], which is closely linked to their ability to modulate phytohormone signaling pathways, as also recently shown for the ABA signaling network [111,112]. As already reported, this signaling ability is deeply dependent on the capacity of flavonoids to regulate “key downstream” components of phytohormone signaling, such as H_2_O_2_ and a range of protein kinases, including PID and mitogen-activated protein kinases (MAPKs) [30,110]. In general, this supports the idea of a primary function of flavonoids as developmental regulators in both the process of plant terrestrialization and the subsequent steps toward drier habitats.

## 4. From UV-B Protection to Biotic Defense: Flavonoids Were Multifunctional from the Very Beginning Assisting the Plant Adaptation to Land

Our analysis leads us to hypothesize that not the increased UV-B irradiance, but the nutrient and water scarcities were the primary environmental drivers for the rise of flavonoid metabolism in substitution of MAAs during plant terrestrialization [48]. This view is in line with the previous hypothesis proposed by Stafford [113]: a nascent flavonoid metabolism unlikely to produce enough flavonoid concentration to constitute an effective shield against the penetration of highly energetic solar wavelengths (i.e., UV-B). Actually, the observation that MAAs have higher molar extinction coefficients than flavonoids over the 290–320 nm portion of the solar spectrum adds strong support to this hypothesis [52]. Indeed, MAAs especially play a pivotal role as “natural sunscreen compounds”, and the UV-B radiation is described as the strongest inducer of their biosynthesis [52], making these compounds excellent candidates as UV-B filters. In addition, the idea that the biosynthesis of flavonoids had not the primary function of providing a more effective shield against UV-B radiation stems from the observation that the spectrum for the induction of flavonoids biosynthesis does not overlap with that of flavonoids absorption [64]. In fact, most flavonoids have maximum absorption peaks between 335 and 360 nm, i.e., in the UV-A portion of the solar spectrum [62,110], whose reception and signaling pathway was already present in algae 1.1 billion years ago [5]. The maximum absorption in these wavelengths is the case, for example, of apigenin and luteolin derivatives found in the liverwort *Marchantia polymorpha* and of quercetin derivatives present in *Physcomitrium* (previously *Physcomitrella*) [114,115,116]. However, it is noteworthy that the reports of the presence of flavonoids in early land plants lineages such as in liverworts and bryophytes did not consider their inter- and intracellular distributions [113], thus making it difficult to conclusively assess the actual significance of these molecules as UV-B screeners. In addition, even previous findings of the induction of biosynthesis of dihydroxy B-ring-substituted flavonoids over that of monohydroxy B-ring-substituted structures in liverworts [114,117], mosses [115,118], and in most angiosperms (for review articles, see [37,119,120]) exposed to UV-B radiation, do not necessarily support the hypothesis of the primary UV-B screening functions of these compounds during the land conquest. Indeed, the maximum absorption of dihydroxy B-ring flavonoids occurs at longer wavelengths than that of monohydroxy B-ring ones. As a whole, this suggests that flavonoids, similarly to other polyphenolic structures, contribute to effectively filtering the UV-B radiation reaching the plant, but it is unlikely that this has been their primary function for the successful adaptation of plants in the newly conquered harsh terrestrial environment.

As outlined in the previous sections, early plants were challenged against the concomitant action of multiple “new” environmental pressures when moving from water to land, high UV-B radiation being just one of these stress agents [55]. The inevitable stress-induced decline in the use of solar irradiation for photosynthesis exposed primitive plants to photooxidative stress, which severity depended only partially on light irradiance. Flavonoids are well known as effective scavengers of reactive oxygen species (ROS), and the steep increase in dihydroxy/monohydroxy B-ring-substituted flavonoids commonly observed in response to UV radiation and high visible light in angiosperms assigns these molecules a pivotal role as antioxidants in the photoprotection process [37,121,122,123]. Notably, increased luteolin-to-apigenin ratio strongly correlated to UV-B irradiance in *M. polymorpha* [114], and the exclusive accumulation of quercetin in UV-B-treated *P. patens* [115] also offers evidence of this pivotal antioxidant role of flavonoids during photoprotection, not only in modern but in early land plants as well. This is remarkable and consistent with the recent observation that the UV-B signaling pathway, mediated by the UVR8 photoreceptor and the downstream *HY5* gene, is conserved in liverworts, mosses, and angiosperms [117,124]. Remarkably, UVR8-induced activation of *HY5* enhances the expression of transcriptional regulators of flavonol biosynthetic genes (e.g., *MYB11*, *MYB12*, and *MYB13*) [125,126,127], the ancient class of flavonoids with the greatest antioxidant potential [48,120]. Moreover, the fact that *HY5* and its homolog *HYH* regulate not only the UV-B responses but also those triggered by blue and red light through flavonoid accumulation [128] reinforces the action of these compounds as much more than only UV-B filters during photoprotection (Figure 2b).

The location of flavonoids in mesophyll cells, in their chloroplasts, and even in the nuclei [37,120] also supports their more prominent action as antioxidant photo-protectants. While flavonoids are in high concentrations and optimally located in subcellular compartments prone to oxidative stress in current-day land plants [37,45,111,112,120], it is still uncertain their actual concentration in early land plants. This makes it complex to assess the actual significance of flavonoids as antioxidants in bryophytes. Nonetheless, we observe that flavonoids in low μM concentrations are enough to keep the ROS at sub-lethal levels [118,129]. This leads to hypothesize that the antioxidant functions of flavonoids in photoprotection increased in significance during the evolution of their metabolism, in turn allowing land plants to radiate toward more stressful habitats. As already noted [30,48,110,130], the same physico-chemical features conferring ROS scavenging ability to flavonoids also allow them to modulate the phytohormones signaling pathways, such as the auxin signaling (and ABA as well [111,112]). Certain flavonoids, especially the antioxidant flavonol quercetin, are particularly effective in modulating both the transport of auxin (by acting as PAT inhibitor) and the auxin signaling pathway through their ROS scavenging abilities [48,130]. Antioxidant flavonoids are also the most effective in regulating plant architecture, acting at root and shoot levels, greatly contributing to the stress-induced morphogenic responses, the “flight” strategy of sessile organisms [131,132,133,134]. We hypothesize that the close auxin-flavonoid relationship ultimately enhanced the morphological complexity of plants at different scales and, hence, improved the capacity of plants to adapt to the harsh terrestrial habitat [30,130].

When moving on land, plants were also challenged against a new set of biotic stress agents [135,136]. Indeed, the emergence of land plants was contemporary to the diversification of some animals and microbes, indicating that the conquest of the terrestrial environment was also marked by novel interactions between plants, symbiotic organisms, and enemies as well [137]. Flavonoids, different from the MAAs, have the capacity to protect the plants from insects, herbivores, and phytopathogens [138]. Therefore, the replacement of MAAs with carbon-based “guard” compounds (i.e., flavonoids) equipped the early land plants with a more versatile arsenal of “defense compounds” sensu lato (Figure 2b). The previous hypothesis that the evolution of the secondary metabolites biosynthetic pathways was for providing defense against natural enemies has been criticized in many instances (for review articles, see [139,140]). Nonetheless, the novel environmental threat imposed by the contemporary presence of insects, herbivores, and pathogens was one of the earliest selective pressures that shaped the evolution of the flavonoid biosynthetic pathway [141,142]. Flavonoids act as both toxic molecules for pathogens and reinforcing compounds of the cell wall structure, thereby reducing the pathogen penetration [136,143]. These compounds also reduce insect oviposition and feeding by decreasing plant tissue palatability while increasing the resistance against pathogenic bacteria and fungi [143]. Recent findings of flavonoid-like pigments being involved in resistance to fungi infection in liverworts suggest the extraordinary ecological value of the evolution of this defense mechanism based on flavonoids [41,142]. The relationship between structure and defense activity of flavonoids has not been analyzed in detail, but the diverse structures may operate well against a wide range of plant natural enemies [62,144,145]. Agati and Tattini [119] hypothesized that angiosperms display a range of flavonoids, usually located in epidermal leaf cells, which are not responsive to light stress, therefore acting probably as a constitutive barrier against both UV radiation and pathogens/predators. This may have also occurred in most bryophytes, which also display a vast arsenal of flavonoid structures with a great defense potential against abiotic and biotic stressors [42,146].

We conclude that the replacement of MAAs with flavonoids during water-to-land transition equipped basal land plants with compounds capable of serving multifarious roles in response to the novel abiotic and biotic challenges imposed by the terrestrial habitat. Although the ability to perform more than one role is a common feature of most secondary/specialized metabolites, which follows nature’s strategy to catch as many flights with one clamp as possible, this is particularly relevant for flavonoids. Even though the evolution of flavonoid metabolism has probably expanded its functional roles in plant-environment interactions, we infer flavonoids have probably been multifunctional since the first appearance of plants on land.

## 5. Conclusions

Plants were challenged against a wide range of novel environmental pressures when moving to land. The excellent capacity of plants to adapt to the harsh terrestrial habitat has depended on both the evolution of a pre-existing molecular toolkit and the rise of a huge number of specialized metabolites. Among the latter, the replacement of MAAs with flavonoids was an extraordinary molecular innovation, producing more than 9000 different flavonoid structures known to date. Modern land plants are capable of finely modulating flavonoid biosynthesis depending on the type and intensity of environmental stressors. In addition, the occurrence of flavonoids in different tissues and subcellular compartments well explains their ability to play multiple functions in response to environmental stimuli. In this review, we reasoned on the multiplicity of roles served by a nascent flavonoid metabolism during the “early steps” of plants in a land environment. Our reasoning stems from the notion that multi-functionality is a widespread property of most specialized metabolites, particularly flavonoids. Although these molecules undoubtedly allowed primitive plants to effectively deal with the most damaging solar wavelengths by acting as both UV filters and ROS scavengers, it is unlikely that these were their most prominent function in the process of terrestrialization. Indeed, we discussed that the benefits associated with the replacement of MAAs (excellent UV filters) with flavonoids mostly have involved the ability of these last to modulate phytohormone signaling; to assist the plant-fungus symbiosis, which is essential to plant development in the “new world” scarce of water and nutrients; besides offering effective defense against pest and pathogens.

Recent studies on genome sequencing of both streptophyte algae and basal land plants (e.g., *M. polymorpha* and *P. patens*) have greatly advanced our knowledge about the molecular features of the evolution and the rise of specialized metabolic pathways that accompanied the plant terrestrialization. Despite the vast number of studies focused on angiosperms, very few have explored the effects of abiotic and biotic stressors on the biosynthesis of flavonoids in bryophytes, the earliest land plant group. Consequently, the functional roles of flavonoids in the interactions between early terrestrial plants and the environment often extrapolate from studies conducted on more complex plants. However, both the concentration in plant tissues and the inter- and intracellular location of flavonoids depends upon the evolution of the metabolism of these compounds and on the complexity of different land plants as well. This makes it difficult to assess the relative significance of the multifarious functions of flavonoids not only during the water-to-land transition but also in the success of plants in the conquest of strikingly different habitats. Therefore, this matter is critically important and merits further investigation.

## Figures and Tables

**Figure 1 ijms-23-05284-f001:**
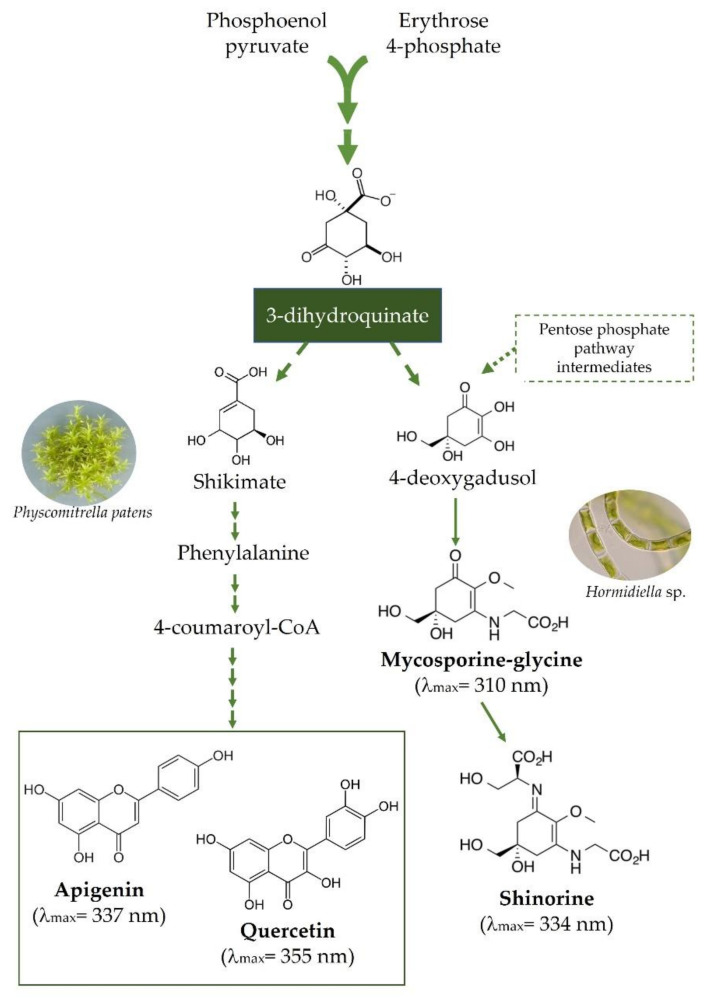
A synthetic diagram showing the metabolic pathways of mycosporine-like amino acids (MAAs, here exampled as the mycosporyne glycine and shinorine) and flavonoids (here represented by the flavone apigenin and the flavonol quercetin). The two pathways diverge early in the shikimate pathway. Mycosporyne glycine and shinorine have, on average, molar extinction coefficients 80% higher than those of apigenin and quercetin over the UV-B portion of the solar spectrum (280–315 nm, G. Agati personal communication).

**Figure 2 ijms-23-05284-f002:**
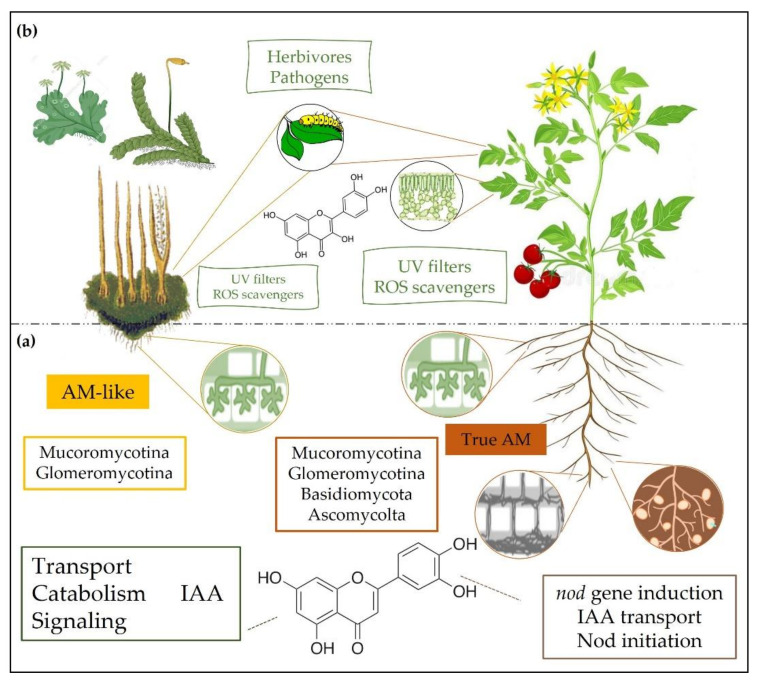
Flavonoids played multiple functions during the establishment of plants on land and their subsequent spread to harsh terrestrial habitats. (**a**) We hypothesize that the successful adaptation of plants on soils largely deficient in water and nutrients was favored by the arbuscular mycorrhiza-like association (AM-like) between early bryophytes and both Glomeromycota and Mucoromycota fungi, assisted by flavonoids. Flavonoids primarily act in modulating auxin transport, catabolism, and hence IAA-signaling during AM-like establishment. Arbuscular mycorrhiza (AM) is an ancient robust trait of land plants since >80% of extant land plants form true AM with a wide range of AM fungi (Glomeromycota, Mucoromycota, Basidiomycota, and Ascomycota). Similar action modes well explain the essential roles of flavonoids in the nodulation observed in a range of angiosperms since these compounds act in Nod initiation by inducting *nod* genes, besides acting in IAA transport. (**b**) Flavonoids also played pivotal roles in early land plants exposed to a novel set of stressful agents associated with the terrestrial habitat, of both abiotic and biotic origins. These functions include the screening of the shortest (UV) solar wavelengths, the scavenging of high light-induced reactive oxygen species (ROS), and the protection against a new set of pathogens and predators. Here we have reasoned that absorption of UV-B radiation and ROS scavenging increased in significance following the evolution of flavonoid metabolism and the increased complexity of land plants.

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
