# Peer review of "Beyond Photoprotection: The Multifarious Roles of Flavonoids in Plant Terrestrialization"

_ijms, 2022, doi:10.3390/ijms23095284_

Round 1
Reviewer 1 Report
This is an interesting review that discusses the multifarious roles of flavonoids in plant terrestrialization besides photoprotection highlighting their part in nutrient acquisition, phytohormone regulation and biotic stress defense. This is certainly riveting from an evolutionarily biochemistry perspective. My only comment is that this paper might be benefit from visualization. For example, L91 An illustration that may include general mycosporine like aminoacds (MAA) metabolism and flavonoid biosynthetic pathway would be helpful for readers to get the idea of ‘replacement’. L105 It would be great to show some skeletons of classical MAA and flavonoids. etc.
Overall, a nice piece of review is impressed not only by the idea but also by the illustration of the conclusion.
Author Response
We thank the reviewer for the comments. The concerns raised by the reviewer are correct. We have revised the MS adding the figures (Fig.1 and Fig.2), as requested. We believe that both figures will offer the readers a more comprehensive view of the subject.
Reviewer 2 Report
The present manuscript comprehensively reviewed the multifarious roles of flavonoids in 2 plant terrestrialization. However, the manuscript lacks pictorial illustration (see the comments annotated on pdf file).

Author Response
We thank the reviewer for the comments.
The present manuscript comprehensively reviewed the multifarious roles of flavonoids in plant terrestrialization. However, the manuscript lacks pictorial illustration (see the comments annotated on pdf file).
Add a related figure depicting
A pictorial illustration of AM association as aided by flavonoids should be added
Add a figure highlighting the role of flavonoids in plant defence. Also add a table related to this section
Answer: We really do not understand the comment given by the colleague about the Table. Instead, we fully agree that a picture illustrating the multifarious roles of flavonoids in assisting the adaptation of early plants to the novel environmental pressures associate to the terrestrial habitat (as well as the adaptation of current day terrestrial plants to the wide range of abiotic and biotic stressors) will improve the MS quality. The figure addressing both suggestions was included as Fig.2.
Add knowledge gaps future research directions
Answer: The knowledge gaps and the directions for further investigations were added in the conclusion (l.367-380).
Round 2
Reviewer 2 Report
I have now reviewed the revised version of the manuscript entitled "Beyond photoprotection: the multifarious roles of flavonoids in 2 plant terrestrialization". The current manuscript is much improved and author has incorporated most of the suggested changes. However, the author should include a paragraph on future research directions at the end of conclusion section. A table on section 3 and 4 discussing mycorrhizal association and UV protection to biotic stress tolerance by flavanoids would be more interesting